# Nonlinear thresholds in lipid-frailty interplay: Precision targets for severe airflow limitation in aging adults

Shuang Deng[1], Zhongqiang Guo[2]*

1 School of Health, Tianhua College, Shanghai Normal University, Shanghai, China, 2 School of Nursing and Health, Henan University, Kaifeng, Henan, China

* guozhongqiang0701@gmail.com

## Abstract

### Objective

To explore the association between fat metabolism, frailty phenotype, and Severe Airflow Limitation（SAL）in middle-aged and elderly populations, identify non-linear thresholds and sociodemographic modification effects.

### Method

This cross-sectional study included 2,907 participants (556 SAL cases) from the China Health and Retirement Longitudinal Study (CHARLS). Associations between lipid indices (AIP, residual cholesterol, etc.), frailty index, and SAL were examined using multivariate logistic regression. Nonlinear relationships were assessed using piecewise regression with the segmented package to identify thresholds. Subgroup and interaction analyses were conducted to evaluate effect modifications by age, gender, education, and other factors.

### Result

AIP showed an inverse association with SAL (fully adjusted OR = 0.556, 95% CI: 0.394–0.787; $P < 0.001$). Residual cholesterol exhibited a nonlinear association with a threshold at 0.329 mmol/L: below this threshold, the inverse association was substantially stronger (OR = 0.003, 95% CI: 0.000–0.473; $P = 0.024$); above the threshold, the inverse association was attenuated but remained significant (OR = 0.758, 95% CI: 0.595–0.967; $P = 0.026$). Frailty status was positively associated with SAL (OR = 1.816, 95% CI: 1.467–2.248; $P < 0.001$). Education level modified the associations of AIP (interaction $P = 0.036$) and residual cholesterol (interaction $P = 0.009$), with the strongest inverse associations observed in the highest education group. Rural residents had a higher prevalence of SAL than urban residents (74.8% vs. 69.3%, $P = 0.010$).

**Data availability statement:** 1. Raw data:https://doi.org/10.6084/m9.figshare.31500256 2.README:https://doi.org/10.6084/m9.figshare.31500391 3.Figure 1. Flow diagram for participants included in the study:https://doi.org/10.6084/m9.figshare.31501363 4.Figure 2. Smooth Curve Fitting:https://doi.org/10.6084/m9.figshare.31501618 5.Figure 3. Subgroup analysis:https://doi.org/10.6084/m9.figshare.31501681 6.Table 1. Baseline characteristics of population stratified by SAL (N=2907):https://doi.org/10.6084/m9.figshare.31501195 7.Table 2. Multiple logistic regression analyses for SAL:https://doi.org/10.6084/m9.figshare.31501246 8.Table 3. Threshold Effect Analysis:https://doi.org/10.6084/m9.figshare.31501294 9.Table 4. Interaction Analysis:https://doi.org/10.6084/m9.figshare.31501339 10.Supplementary Table 1. FDRadjusted results for multivariate logistic regression analyses:https://doi.org/10.6084/m9.figshare.31500484 11.Supplementary Table 2. FDRadjusted results for threshold effect analyses:https://doi.org/10.6084/m9.figshare.31500586 12.Supplementary Table 3. FDRadjusted results for interaction analyses:https://doi.org/10.6084/m9.figshare.31500646 13.Supplementary Table 4. Bootstrap validation of threshold estimates:https://doi.org/10.6084/m9.figshare.31500694 14.Supplementary Table 5. Sensitivity analysis excluding the highededucation subgroup:https://doi.org/10.6084/m9.figshare.31500739 15.Supplementary Table 6. Sensitivity analysis using multiple imputation:https://doi.org/10.6084/m9.figshare.31500766 16.Supplementary Table 7. Multivariable logistic regression analyses after excluding outliers identified by the IQR method:https://doi.org/10.6084/m9.figshare.31500793 17.S2 Supplementary Table 8: Full multivariate logistic regression results for all lipid parameters, muscle indices, and socioeconomic variables:https://doi.org/10.6084/m9.figshare.31891360.v1 18.S2 Supplementary Table 9: Full threshold analysis results for all continuous exposures:https://doi.org/10.6084/m9.figshare.31891360.v1 19.S2 Supplementary Table 10: Full interaction analysis results for all exposures stratified by education level:https://doi.org/10.6084/m9.figshare.31891360.v1 20.Supplementary Code 1. R code for data processing and outlier removal:https://doi.org/10.6084/m9.figshare.31500859 21.Supplementary

## Conclusion

In this cross-sectional study, lower AIP and residual cholesterol levels below 0.329 mmol/L, as well as frailty, were associated with SAL, particularly among older and rural populations. The observed lipid profiles may reflect disease-related metabolic alterations. The modification of lipid-SAL associations by education level suggests that social factors may be relevant for identifying high-risk populations.

## Introduction

Chronic respiratory diseases (CRDs), including chronic obstructive pulmonary disease and interstitial lung disease, are one of the leading causes of disability and death worldwide. According to data from 2019, these diseases rank as the third leading cause of death worldwide, causing approximately 4 million deaths and affecting 454.6 million patients, with significantly higher disease burdens observed in low-income countries.Its pathological mechanism involves complex interactions of multidimensional factors such as genetics, environment, and metabolism [1,2]. Although traditional risk factors (such as smoking and air pollution) have been widely recognized as the main driving factors (smoking accounts for 46.0% of CRD attributable deaths, and air pollution accounts for 38.2%) [1,3], the role of lipid metabolism disorder (characterized by atherosclerosis index AIP, visceral adiposity index VAI) and somatic asthenia syndrome (frailty sarcopenia axis) in the progressive decline of lung function has not been clear.

The Visceral Adiposity index (VAI) is calculated based on waist circumference, body mass index (BMI), and gender-specific indicators constructed by triglycerides and HDL cholesterol can indirectly reflect visceral fat dysfunction and insulin resistance (Rs = −0.721, P < 0.001) [4], and they are independently associated with the risk of cardiovascular and cerebrovascular events (OR=2.45).Especially in middle-aged and elderly populations, where such metabolic imbalances may exacerbate airway inflammation and lung tissue remodeling through a pro-inflammatory microenvironment [5], while muscle mass reduction (such as a decrease in the skeletal muscle mass index (ASM), directly impairs respiratory mechanical efficiency – studies have found that ASM is significantly positively correlated with lung capacity and maximum expiratory pressure ($P < 0.001$) [6,7], and the lung function (FEV1/FVC) and exercise tolerance (6MWD) of weakened patients significantly deteriorate [8].These two mechanisms work synergistically to accelerate the development of Severe Airflow Limitation (SAL) [9].

In recent years, the impact of Social Determinants of Health (SDH) on respiratory health has gradually received attention.Research has shown that low education levels and rural living environments may indirectly regulate the aforementioned biological processes through differences in health literacy, accessibility to medical resources, and behavioral risks such as smoking and nutritional imbalances.For example, African American children have an increased risk of asthma attacks due to a lack of medical resources [10], and a negative SDH status reduces the survival rate of advanced lung

Code 2. R code for multiple imputation:https://doi.org/10.6084/m9.figshare.31500922

22.Supplementary Code 3. R code for VIF calculation and FDR correction: https://doi.org/10.6084/m9.figshare.31500991.

**Funding:** The author(s) received no specific funding for this work.

**Competing interests:** The authors have declared that no competing interests exist.

cancer patients by 24% (aHR = 1.24) [11].However, existing literature lacks a quantitative analysis of the interaction between biomarkers and social factors. A scoping review revealed that only 0.3% of respiratory rehabilitation studies reported social class variables [12]and did not clarify whether these relationships exhibit non-linear threshold effects, such as the "critical turning point" of VAI on lung function [15].

Based on this, this study utilized the 2015 China Health and Retirement Longitudinal Study (CHARLS) national cohort (N = 2907), which covered 17500 individuals aged 45 and above in 150 districts and counties across 28 provinces using multistage stratified sampling. The data integrated socioeconomic, biomarker, and clinical indicators [13,14]. For the first time, it combined multidimensional variables such as the Atherosclerotic index of Plasma (AIP), Visceral Adipose index, VAI, Residual Cholesterol (RC), Frailty index, Accessory Skeletal Muscle Mass (ASM), and Social Isolation to address the key question of whether lipid metabolism and frailty independently predict SAL risk?Based on the synergistic mechanism hypothesis of VAI metabolic disorder and ASM decline [4,7], do social factors, such as education level and residential area, modify the effects of the above biomarkers?(Response to the lack of evidence on SDH regulation of respiratory health) [12].

Is there a non-linear correlation between key continuous variables, such as VAI, frailty index, and SAL? What is the threshold point?Drawing on the non-linear method of identifying biomarker inflection points using the Sigmoid function model [15], the findings of this study will provide a novel bio social integrated prediction model for the early screening of SAL and evidence-based respiratory health intervention strategies for low educated populations.

## Materials and methods

### Study population

This study is based on publicly available data from the China Health and Retirement Longitudinal Study (CHARLS) database. The final sample size included in the analysis was 2907 participants, including 2351 in the group without Severe Airflow Limitation and 556 in the group with Severe Airflow Limitation. Participants were included in this analysis if they had complete data on the key exposure variables (lipid parameters, frailty index, and ASM) and the outcome variable (PEF% predicted). Participants with missing data on covariates were handled using multiple imputation as described in the Statistical Analysis section. No additional inclusion or exclusion criteria beyond data availability were applied to maximize sample representativeness. The study detailed the baseline demographic characteristics of participants (such as age, gender, place of residence, marital status, education level) and lifestyle information (such as smoking history, frequency of alcohol consumption) [16].All analyses were conducted based on the available variables in the database.This study used publicly available data from the 2015 nationwide follow-up wave of the China Health and Retirement Longitudinal Study (CHARLS).The data was obtained through authorization from the CHARLS Data Center of Peking University on May 19, 2025.The original CHARLS study was approved by the Biomedical Ethics Review Committee of

Peking University (IRB00001052−1015). All participants in the original CHARLS survey provided written informed consent prior to data collection. For this secondary analysis, all data were fully anonymized and de-identified prior to receipt by the authors. As this study used only publicly available, de-identified data from the CHARLS database, no additional ethical approval or informed consent was required [17,18] (Fig 1).

## Variable definition and measurement

**Outcome variable.** Severe Airflow Limitation（SAL）: Diagnostic criteria: PEF% pred<60% is defined as severe airflow limitation.Predicted PEF for males: $75.6 + 20.4 \times age − 0.41 \times age^2 + 0.002 \times age^3 + 1.19 \times height$ (cm) Predicted PEF for females : $282.0 + 1.79 \times age − 0.046 \times age^2 + 0.68 \times height(cm)$
PEF% pred = (measured PEF value/predicted PEF value) × 100%. Severe Airflow Limitation (SAL) was defined as PEF% predicted<60% [19,20]. Importantly, this definition is not equivalent to the GOLD standard diagnosis of airflow obstruction, which requires spirometry-confirmed FEV1/FVC<0.70. PEF reflects large airway function but cannot distinguish obstructive from restrictive patterns. Therefore, our outcome should be interpreted as 'severe PEF reduction' rather than clinically confirmed airflow obstruction.

## Core independent variable

**Lipid metabolism indicators.** Atherogenic index of Plasma, AIP: $AIP = \log_{10}\left(\frac{Triglycerides}{HDL−C}\right)$, Reflect the potential of lipid induced atherosclerosis,where both triglycerides and HDL-C are in mmol/L [21].

$$Visceral\ Adiposity\ index,\ VAI : VAI_{men} = \left(\frac{Waist\ circumference}{39.68 + (1.88 \times BMI)}\right) \times \left(\frac{Triglycerides}{1.03}\right) \times \left(\frac{1.31}{HDL − C}\right)$$

$VAI_{women} = \left(\frac{Waist\ circumference}{36.58+(1.89 \times BMI)}\right) \times \left(\frac{Triglycerides}{0.81}\right) \times \left(\frac{1.52}{HDL−C}\right)$,Indicators reflecting visceral fat function and quantity calculated based on waist circumference, body mass index, triglycerides, and high-density lipoprotein cholesterol,(All lipid units in mmol/L) [22].

Non-High-Density Lipoprotein Cholesterol, NHDL: NHDL = Total cholesterol − HDL − C, Represents the total cholesterol contained in all atherogenic lipoprotein particles,with both in mmol/L. [23].

Residual Cholesterol,RC: RC = Total cholesterol − HDL−C − LDL−CRepresents cholesterol components rich in tri-glycerides and lipoproteins,with all components in mmol/L. [24].

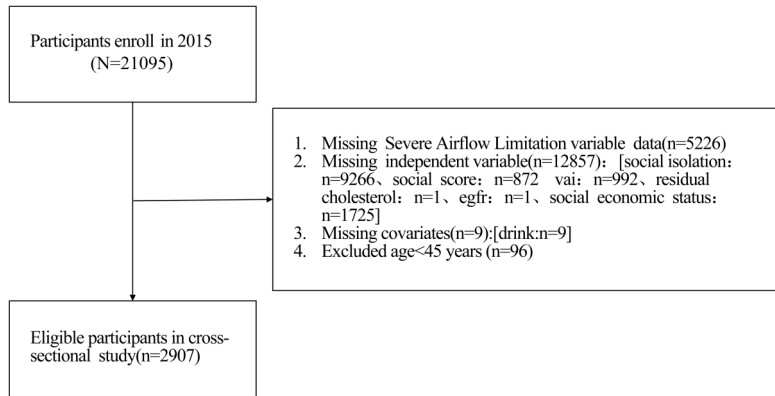

**Fig 1. Flow diagram for participants included in the study.**

## Frailty and muscle indicators

Frailty index: $\text{Frailty index} = \frac{\text{Number of health deficits present}}{\text{Total number of deficits assessed}}$, Comprehensive assessment of individual physiological reserve decline and vulnerability [25].

Frailty:Binary variable based on frailty index definition (Yes/No) [26].

Accessory Skeletal Muscle Mass, ASM:

$\text{ASM} = 0.193 \times \text{weight(kg)} + 0.107 \times \text{height(cm)} - 4.157 \times \text{gender} - 0.037 \times \text{age} - 2.631$ Male gender = 1, female = 2, Sum of skeletal muscle mass in arms and legs(kg), Assess muscle mass [27, 28, 29].

$$\text{Castelli index I} : \text{Castelli Index I} = \frac{\text{Total cholesterol}}{\text{HDL} - \text{C}}$$

$$\text{Castelli index II} : \text{Castelli Index II} = \frac{\text{LDL} - \text{C}}{\text{HDL} - \text{C}}$$

Estimated Glomerular Filtration Rate, eGFR.

Female:

- $\text{Scr} \leq 62\ \mu\text{mol/L} \rightarrow \text{eGFR} = 144 \times (\text{Scr}/62)^{-0.329} \times 0.993\text{Age}$

- $\text{Scr} > 62\ \mu\text{mol/L} \rightarrow \text{eGFR} = 144 \times (\text{Scr}/62)^{-1.209} \times 0.993\text{Age}$

Male:

- $\text{Scr} \leq 80\ \mu\text{mol/L} \rightarrow \text{eGFR} = 141 \times (\text{Scr}/80)^{-0.411} \times 0.993\text{Age}$

- $\text{Scr} > 80\ \mu\text{mol/L} \rightarrow \text{eGFR} = 141 \times (\text{Scr}/80)^{-1.209} \times 0.993\text{Age}$

, Assess renal function [30].

## Social factors

Social isolation: defined based on the frequency of social activities (≥ 2 points considered as isolation) [31].

Socioeconomic status: a composite indicator constructed by comprehensive income, occupation, and education,SES is divided into four categories: low/low (score 0), medium low/low middle (score 1 or 2), medium high/puppet middle (score 3 or 4), and high/high (score 5) [32].

## Covariant

Demographic factors: age (binary classification: ≥ 60 years/ < 60 years) [33], gender [34].

Behavioral factors: smoking history (NO/YES) [35], drinking frequency (Low frequency/Intermediate frequency/High frequency) [36].

Social background: place of residence (urban/rural) [37], marital status (Unmarried/Married) [38], education level (No formal education/high school and below/Above high school) [39].

## Statistical analysis

Adopting a four stage modeling process (software: R 4.0 and EmpowerStats 6.0) [40–42]:

Normality of continuous variables was assessed using the Shapiro-Wilk test and visual inspection of histograms. Variables with approximately normal distribution are presented as mean ± SD and compared using independent t-tests. Skewed variables are presented as median (interquartile range, IQR) and compared using Mann-Whitney U tests.

Multivariate logistic regression:Basic Model: Age and Gender Correction.Fully corrected model: additional correction for place of residence, marriage, education, smoking, and alcohol consumption.Result report: odds ratio (OR) and 95% confidence interval (95% CI).

Nonlinear relationship test: To identify potential threshold effects, we fitted piecewise logistic regression models using the segmented package in R. For each continuous exposure, a two-piecewise linear model was fitted, and the optimal breakpoint (threshold, K) was estimated via maximum likelihood methods. Nonlinearity was assessed by comparing the one-line (linear) model with the two-piecewise model using a log-likelihood ratio test (P<0.05 indicating significant non-linearity). These analyses were exploratory and hypothesis-generating; thresholds were not pre-specified based on prior literature.

Stratification and interaction analysis:Modeling by age, gender, education level, etc.

Interaction test: Likelihood ratio test compares nested models with/without interaction terms.

The complete R analysis code used for data processing, imputation, VIF calculation, and FDR correction is provided as Supplementary Code 1–3.

## Result

### Baseline characteristics

The study population comprised 2,907 participants, stratified into those without SAL (n=2,351, 80.9%) and with SAL (n=556, 19.1%). Robust differences were observed between the groups. Participants with SAL were significantly older (≥60 years: 67.6% vs. 57.3%, $P<0.001$), more likely to reside in rural villages (74.8% vs. 69.3%, $P=0.010$), less educated (no formal education: 53.6% vs. 41.7%, $P<0.001$), more often unmarried (28.1% vs. 21.7%, $P=0.001$), and reported higher social isolation (44.2% vs. 35.9%, $P<0.001$). Frailty prevalence was substantially higher in the SAL group (31.1% vs. 18.7%, $P<0.001$), reflected in a greater mean frailty index (6.694±4.529 vs. 5.118±3.867, $P<0.001$). Socioeconomically, the SAL group had a higher proportion in the low SES category (32.6% vs. 26.4%, $P=0.005$). Significant metabolic differences included lower values in the SAL group for VAI (4.473±3.990 vs. 4.994±4.420, $P=0.011$), AIP (0.340±0.276 vs. 0.392±0.283, $P<0.001$), non-HDL cholesterol (77.472±85.888 vs. 92.501±95.316, $P<0.001$), residual cholesterol (0.727±0.392 vs. 0.792±0.465, $P=0.002$), and eGFR (87.385±17.024 vs. 89.472±15.890, $P=0.006$). Muscle mass indices (ASM, Castelli index I, Castelli index II) were also significantly lower in the SAL group (all $P<0.001$). No significant differences were found for gender, smoking status, or drinking frequency (all $P>0.05$).The lack of significant differences in these variables between groups may reflect disease-related behavioral changes—individuals with severe airflow limitation may reduce or quit smoking and alcohol consumption due to worsening respiratory symptoms—as well as survival bias in this older population, where higher mortality among males with severe respiratory disease may balance sex distribution. Multivariate regression analyses were therefore conducted to appropriately adjust for these factors. Full details are presented in Table 1.

### Multivariate regression analysis

Metabolic markers were significantly associated with SAL in the fully adjusted models: lower AIP (OR = 0.556, 95% CI: 0.394–0.787, P<0.001), VAI (OR = 0.974, 95% CI: 0.950–0.998, P=0.033), NHDL (OR = 0.998, 95% CI: 0.997–1.000, P=0.005), and residual cholesterol (OR = 0.732, 95% CI: 0.577–0.927, P=0.010) were each associated with lower odds of SAL. However, given the cross-sectional design, these associations should not be interpreted as causal association (see Discussion). Frailty-related measures showed robust positive associations: frailty diagnosis increased SAL risk by >80% in all models (Adjust II OR=1.816, 95% CI:1.467–2.248, $P<0.001$), while each unit increase in frailty index elevated risk by 8.2% (Adjust II OR=1.082, 95% CI:1.058–1.106, $P<0.001$). Muscle mass indices remained significant protective factors (ASM: Adjust II OR=0.903, $P<0.001$; Castelli 1: OR=0.782, $P<0.001$; Castelli 2: OR=0.729, $P<0.001$). Notably, social isolation

**Table 1. Baseline characteristics of population stratified by SAL (N = 2907).**

| Characteristics | Without SAL | With SAL | *P*-value |
|---|---|---|---|
| **N** | 2351 | 556 | |
| **Age** | | | <0.001 |
| <60 | 1003 (42.663%) | 180 (32.374%) | |
| ≥60 | 1348 (57.337%) | 376 (67.626%) | |
| **Gender** | | | 0.271 |
| Male | 1266 (53.849%) | 285 (51.259%) | |
| Female | 1085 (46.151%) | 271 (48.741%) | |
| **Living Area** | | | 0.010 |
| Urban Community | 722 (30.710%) | 140 (25.180%) | |
| Rural Village | 1629 (69.290%) | 416 (74.820%) | |
| **Married Status** | | | 0.001 |
| Unmarried | 511 (21.735%) | 156 (28.058%) | |
| Married | 1840 (78.265%) | 400 (71.942%) | |
| **Education** | | | <0.001 |
| No formal education | 981 (41.727%) | 298 (53.597%) | |
| High school and below | 1299 (55.253%) | 251 (45.144%) | |
| Above high school | 71 (3.020%) | 7 (1.259%) | |
| **Smoke Status** | | | 0.115 |
| NO | 1212 (51.553%) | 266 (47.842%) | |
| YES | 1139 (48.447%) | 290 (52.158%) | |
| **Drinking Status** | | | 0.273 |
| Low frequency | 1830 (77.839%) | 444 (79.856%) | |
| Intermediate frequency | 152 (6.465%) | 26 (4.676%) | |
| High frequency | 369 (15.695%) | 86 (15.468%) | |
| **Social Isolation** | | | <0.001 |
| NO | 1508 (64.143%) | 310 (55.755%) | |
| YES | 843 (35.857%) | 246 (44.245%) | |
| **Frailty** | | | <0.001 |
| NO | 1911 (81.285%) | 383 (68.885%) | |
| YES | 440 (18.715%) | 173 (31.115%) | |
| **Social Economic Status** | | | 0.005 |
| low | 620 (26.372%) | 181 (32.554%) | |
| low-middle | 1212 (51.553%) | 283 (50.899%) | |
| upper-middle | 505 (21.480%) | 90 (16.187%) | |
| high | 14 (0.595%) | 2 (0.360%) | |
| **VAI** | 3.73 (2.28–6.13) | 3.26 (2.02–5.45) | 0.011† |
| **AIP** | 0.39 ± 0.28 | 0.34 ± 0.28 | <0.001 |
| **NHDL** | 66.61 (29.20–125.59) | 54.02 (21.45–103.03) | <0.001† |
| **Residual Cholesterol** | 0.67 (0.50–0.95) | 0.63 (0.48–0.85) | 0.002† |
| **EGFR** | 89.47 ± 15.89 | 87.39 ± 17.02 | 0.006 |
| **Frailty Index** | 4.32 (2.43–6.79) | 5.64 (3.42–9.50) | <0.001† |
| **ASM** | 17.72 ± 4.16 | 16.48 ± 4.01 | <0.001 |

*(Continued)*

**Table 1.** (Continued)

| Characteristics | Without SAL | With SAL | P-value |
|---|---|---|---|
| Castelli Index I | 3.65 (3.10–4.21) | 3.42 (2.86–4.06) | <0.001† |
| Castelli Index II | 2.08±0.68 | 1.95±0.64 | <0.001 |

Footnote: SAL, Severe Airflow Limitation; VAI, Visceral Adiposity Index (unitless); AIP, Atherogenic Index of Plasma (unitless); NHDL, Non-High-Density Lipoprotein Cholesterol (mmol/L); Residual Cholesterol (mmol/L); eGFR, estimated Glomerular Filtration Rate (mL/min/1.73m²); ASM, Appendicular Skeletal Muscle Mass (kg); Castelli Index I (unitless) = Total cholesterol/HDL-C; Castelli Index II (unitless) = LDL-C/HDL-C. Data are presented as mean±SD for continuous variables and n (%) for categorical variables. P-values were calculated using t-tests for continuous variables and chi-square tests for categorical variables.

† P-values for skewed variables (VAI, NHDL, residual cholesterol, frailty index, Castelli index I) were calculated using Mann-Whitney U test; data are presented as median (interquartile range, Q1–Q3). Normally distributed variables (AIP, eGFR, ASM, Castelli index II) are presented as mean±SD, with P-values from t-test. Categorical variables are presented as n (%).

lost significance after full adjustment (Adjust II OR=1.202, $P=0.115$), and socioeconomic gradients attenuated completely in Adjust II (all $P>0.05$).After applying False Discovery Rate (FDR) correction, all significant associations remained with q<0.05 except for VAI which became suggestive (q=0.043) (Supplementary Table 1). Sensitivity analysis using multiple imputation for missing data yielded consistent estimates (Supplementary Table 6), confirming the robustness of the findings (Table 2).

Full results for all lipid parameters, muscle indices, and socioeconomic variables are provided in S8 Table.

## Threshold effect analysis

Threshold analysis revealed critical nonlinear relationships in SAL predictors (Table 3). Residual cholesterol showed a nonlinear association with a threshold at 0.329 mmol/L: below this threshold, the inverse association was extremely strong (OR = 0.003, 95% CI: 0.000–0.473, P=0.024); above the threshold, the inverse association was attenuated but remained significant (OR = 0.758, 95% CI: 0.595–0.967, P=0.026). VAI showed a J-shaped nonlinear association with SAL, with a turning point at 4.687: below this threshold, lower VAI was associated with lower odds of SAL (OR=0.891, P=0.006); above the threshold, the association was not significant (OR=0.991, P=0.558). Muscle indices also exhibited threshold-dependent associations: below the Castelli index I cutoff (3.567), each unit decrease was associated with 41.8% lower odds of SAL (OR=0.582, P<0.001); below the Castelli index II threshold (1.791), each unit decrease was associated with approximately 50% lower odds (OR=0.498, P<0.001). Above these thresholds, no significant associations were observed. In contrast, AIP (OR=0.556, P=0.001), frailty index (OR=1.077, P<0.001), and ASM (OR=0.881, P<0.001) maintained consistent linear effects. Log-likelihood ratio tests confirmed significant nonlinearity for VAI (P=0.035), residual cholesterol (P=0.039), and both Castelli indices (P≤0.050)(Fig 2). After FDR correction, the nonlinearity for VAI and residual cholesterol became suggestive (q=0.063 and 0.067, respectively), while Castelli index I remained significant (q=0.026) (Supplementary Table 2). Bootstrap validation with 1,000 iterations supported the stability of the thresholds, particularly for residual cholesterol (mean K=0.55, 95% CI: 0.32–1.61) and Castelli index I (mean K=3.48, 95% CI: 2.43–4.18) (Supplementary Table 4).

Full threshold analysis results for all exposures, including VAI, AIP, NHDL, eGFR, frailty index, ASM, and Castelli index II, are provided in Supplementary Table 9.

## Subgroup analysis

Stratified analyses by age group revealed significant modifications in the relationship between cardiometabolic factors and Severe Airflow Limitation. In older adults (n=1,724), visceral adiposity (VAI: OR=0.964, 95%CI:0.932–0.997), atherogenic index (AIP: OR=0.503, 95%CI:0.322–0.786), NHDL cholesterol (OR=0.998, 95%CI:0.996–0.999), and residual cholesterol (OR=0.589, 95%CI:0.423–0.822) demonstrated significant association against pulmonary dysfunction (all $P<0.05$), whereas these associations were non-significant in younger adults (n=1,183). Conversely, frailty indicators consistently elevated risk across both age groups, with frailty index increasing risk by 11.0% (P<0.001) in younger and

**Table 2. Multiple logistic regression equation.**

| Exposure | Non-adjusted | Adjust I | Adjust II |
|---|---|---|---|
| AIP | 0.514 (0.367, 0.720) <0.001 | 0.528 (0.375, 0.743) <0.001 | 0.556 (0.394, 0.787) <0.001 |
| Residual Cholesterol | 0.693 (0.548, 0.877) 0.002 | 0.686 (0.540, 0.873) 0.002 | 0.732 (0.577, 0.927) 0.010 |
| Frailty Index | 1.091 (1.068, 1.114) <0.001 | 1.083 (1.059, 1.107)<0.001 | 1.082 (1.058, 1.106) <0.001 |
| ASM | 0.929 (0.908, 0.951) <0.001 | 0.867 (0.833, 0.901) <0.001 | 0.903 (0.876, 0.932) <0.001 |

Footnote: OR, Odds Ratio; CI, Confidence Interval; AIP, Atherogenic Index of Plasma; NHDL, RC, Residual Cholesterol. Model 1: crude (unadjusted). Model 2: adjusted for age and gender. Model 3: adjusted for age, gender, living area, marital status, education, smoking, and alcohol consumption. All lipid variables were analyzed in separate models.

**Table 3. Threshold effect analysis.**

| exposure: | Residual Cholesterol | Castelli Index I |
|---|---|---|
| Outcome:SAL | OR (95%CI) *P*value | OR (95%CI) *P*value |
| A straight line effect | 0.716 (0.562, 0.912) 0.007 | 0.774 (0.690, 0.869) <0.001 |
| Model II | | |
| Folding point (K) | 0.329 | 3.567 |
| n below K/ n above K | 130/2777 | 1386/1521 |
| <K-segment effect 1 | 0.003 (0.000, 0.473) 0.024 | 0.582 (0.457, 0.742) <0.001 |
| >K-segment effect 2 | 0.758 (0.595, 0.967) 0.0257 | 0.927 (0.793, 1.084) 0.3414 |
| Log Likelihood Ratio Tests | 0.039 | 0.011 |

Footnote: RC, Residual Cholesterol;K, inflection point (threshold); LRT, Likelihood Ratio Test for nonlinearity. ORs and 95% CIs are presented for linear effects and for segments below (<K) and above (>K) the estimated threshold. All analyses were adjusted for age, gender, living area, marital status, education, smoking, and alcohol consumption.n below K and n above K indicate the number of participants in each segment of the piecewise regression model for variables with significant nonlinearity.

6.4% (*P*<0.001) in older adults, while frailty status nearly doubled risk in younger adults (OR=2.258, *P*<0.001). Accessory Skeletal Muscle Mass (ASM) showed significant inverse associations with SAL in both age groups (younger: OR=0.870, *P*<0.001; older: OR=0.885, *P*<0.001). Cardiovascular indices (Castelli I/II) showed stronger association in older adults, particularly Castelli Index I (OR=0.751, *P*<0.001)(Fig 3).

## Interaction analysis

Stratification by educational attainment revealed effect modifications in SAL risk associations (Table 4). The inverse association between AIP and SAL appeared substantially stronger in the highest education group (OR = 0.015, 95% CI: 0.000–0.855, P=0.042) compared to those with no formal education (OR = 0.773, P=0.292; P for interaction=0.036). Similarly, the inverse association for residual cholesterol was observed only among highly educated individuals (OR = 0.009, 95% CI: 0.000–0.709, P=0.035 vs. OR = 0.908, P=0.539 in no formal education; P for interaction=0.009). However, these findings must be interpreted with extreme caution due to the very small sample size in the highest education stratum (n=78 total; only 7 SAL cases), which renders the estimates unstable and likely not replicable. After FDR correction, the AIP×education interaction became suggestive (q=0.085), while the residual cholesterol×education interaction remained significant (q=0.031) (Supplementary Table 3). Sensitivity analysis excluding the high-education subgroup produced estimates nearly identical to the main analysis (Supplementary Table 5), confirming that overall findings were not driven by this small subgroup.

Conversely, the inverse associations for muscle-related indices were stronger in lower education strata: Castelli index I showed a stronger inverse association in the high school education group (OR = 0.728, P<0.001) compared to those

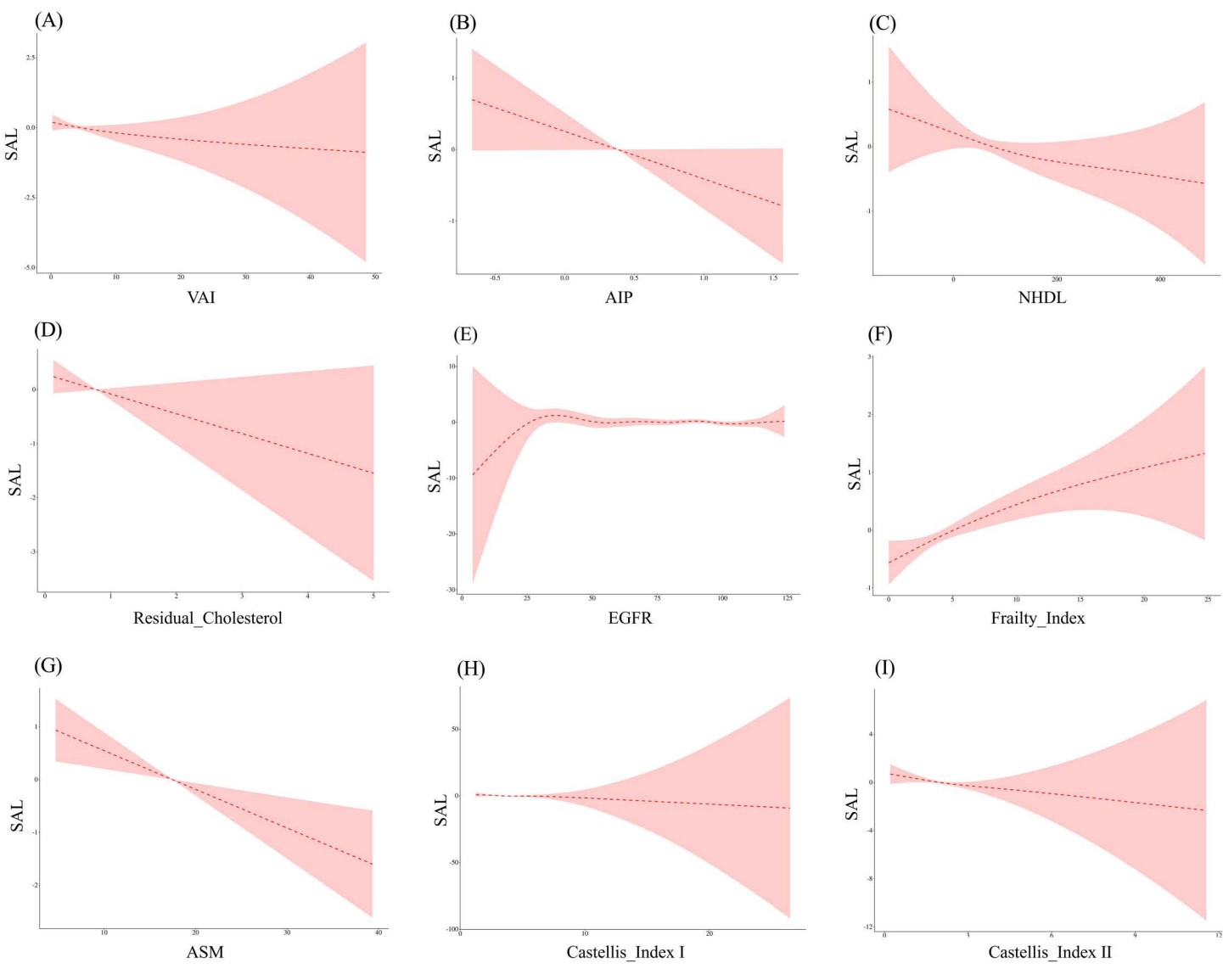

**Fig 2. Smooth curve fitting.**

with no formal education (OR = 0.855, P = 0.051). ASM was significantly associated with lower odds of SAL only in the lower education groups, with no significant association observed in the highest education stratum (OR = 0.984, P = 0.863). NHDL showed a significant inverse association only in the intermediate education group (high school/below: OR = 0.997, P = 0.003; P for interaction = 0.031). Notably, frailty index consistently elevated SAL risk by 6.8–7.3% per unit (P < 0.001) across all education levels without significant interaction (P interaction > 0.05).

## Discussion

An important consideration when interpreting the lipid findings is the strong possibility of reverse causation. The SAL group exhibited significantly lower levels of multiple lipid parameters (AIP, NHDL, RC) compared to controls, yet these lower levels were statistically associated with reduced odds of SAL (OR < 1). This pattern is consistent with the

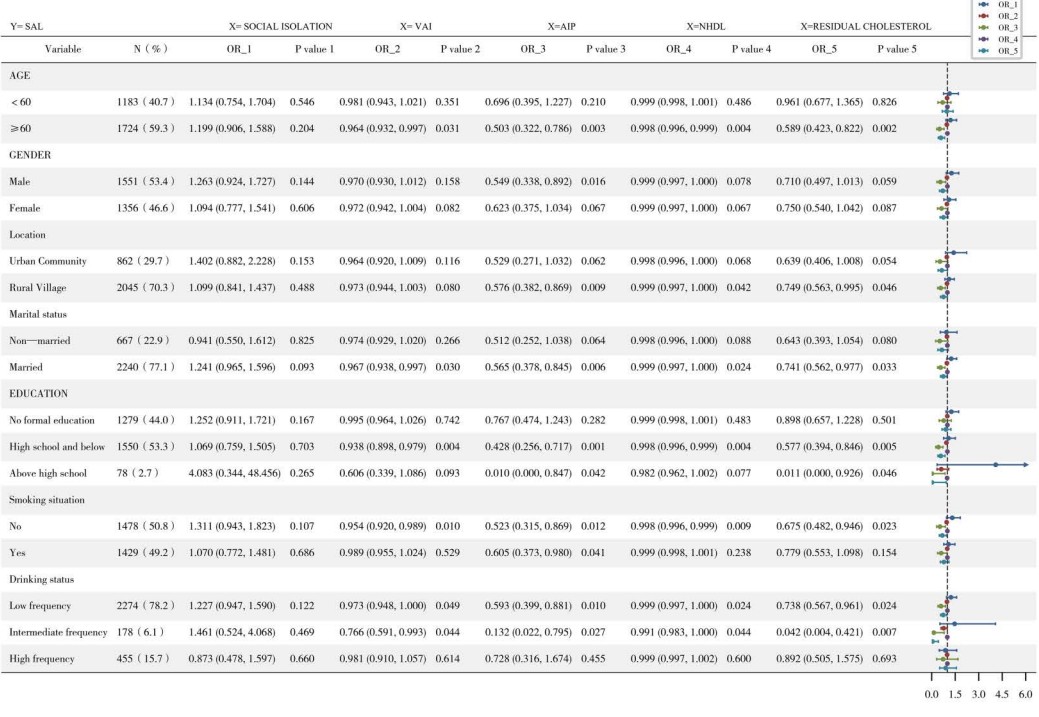

**Fig 3. Subgroup analysis.**

**Table 4. Interaction analysis.**

| Model | No formal education | High school and below | Above high school | P interaction |
|---|---|---|---|---|
| Outcome:SAL | OR (95%CI) Pvalue | OR (95%CI) Pvalue | OR (95%CI) Pvalue | |
| **AIP** | | | | |
| Crude | 0.745 (0.465, 1.194) 0.221 | 0.428 (0.261, 0.704) <0.001 | 0.031 (0.001, 0.957) 0.047 | 0.057 (0.038 #) |
| Model I* | 0.766 (0.476, 1.233) 0.273 | 0.433 (0.262, 0.716) 0.001 | 0.042 (0.001, 1.255) 0.067 | 0.071 (0.039 #) |
| Model II* | 0.773 (0.479, 1.248) 0.292 | 0.427 (0.257, 0.710) 0.001 | 0.015 (0.000, 0.855) 0.042 | 0.036 (0.029 #) |
| **Residual Cholesterol** | | | | |
| Crude | 0.876 (0.644, 1.191) 0.398 | 0.580 (0.400, 0.841) 0.004 | 0.010 (0.000, 0.678) 0.032 | 0.009 (0.019 #) |
| Model I* | 0.893 (0.655, 1.218) 0.476 | 0.571 (0.391, 0.833) 0.004 | 0.013 (0.000, 0.872) 0.043 | 0.012 (0.017 #) |
| Model II* | 0.908 (0.667, 1.236) 0.539 | 0.599 (0.412, 0.870) 0.007 | 0.009 (0.000, 0.709) 0.035 | 0.009 (0.020 #) |

Model I has been adjusted:age, gender and the interaction terms for following variables: age, gender.

Model II has been adjusted:living area, marital status, smoke status, drinking status and the interaction terms for following variables: living area, marital status, smoke status, drinking status.

Sample sizes: No formal education (n = 1,279; SAL = 298), High school and below (n = 1,550; SAL = 251), Above high school (n = 78; SAL = 7).Cell size < 30 participants in the above-high-school stratum (n = 78 total; 7 SAL cases); estimates may be unstable and should be interpreted with caution. These findings are exploratory and hypothesis-generating only, and may not be replicable in larger samples.

well-described cachexia phenotype in chronic lung diseases such as COPD, where systemic inflammation, elevated resting energy expenditure, and tissue wasting lead to depletion of fat mass and circulating lipids [43]. A recent study from the LEAD cohort similarly demonstrated that alterations in body composition, including reduced muscle mass and altered fat distribution, are closely linked to lung function impairment in both adults and elderly populations [44]. These findings collectively suggest that the lower lipids observed in SAL patients are likely a consequence of the disease process rather than protective factors. This interpretation is supported by the consistency of findings across multiple sensitivity analyses, including multiple imputation for missing data (Supplementary Table 6) and exclusion of the small high-education subgroup (Supplementary Table 5), which yielded similar estimates and confirmed that the main results are not driven by missing data or unstable subgroups.

This study identified nonlinear associations between several lipid parameters and SAL. VAI showed a J-shaped relationship with a threshold at 4.687: below this threshold, lower VAI was associated with lower odds of SAL (OR = 0.891, P = 0.006); above the threshold, no significant association was observed. This pattern aligns with previous research demonstrating an inverse relationship between VAI and lung function parameters in the Chinese population [45]. However, our finding of a nonlinear threshold extends this knowledge by suggesting that the relationship between visceral adiposity and lung function may be more complex than previously appreciated, potentially reflecting the dual role of adipose tissue in both metabolic regulation and inflammatory processes [45,46]. The threshold was relatively stable in bootstrap validation (mean K = 4.56, 95% CI: 1.11–12.22; Supplementary Table 4), though the wide confidence interval indicates some uncertainty in the precise estimate.

Similarly, residual cholesterol demonstrated a striking threshold effect at 0.329 mmol/L: below this value, the inverse association was extremely strong (OR = 0.003, 95% CI: 0.000–0.473), while above the threshold, a weaker but significant inverse association persisted (OR = 0.758, 95% CI: 0.595–0.967). Bootstrap validation supported the stability of this threshold (mean K = 0.55, 95% CI: 0.32–1.61; Supplementary Table 4). After applying False Discovery Rate (FDR) correction, the nonlinearity for VAI and residual cholesterol became suggestive (q = 0.063 and 0.067, respectively), while Castelli index I remained significant (q = 0.026) (Supplementary Table 2) [4,47]. These nonlinear patterns may involve pathways distinct from pro-inflammatory mechanisms, though direct investigation of such mechanisms was beyond the scope of this study, and the extremely low residual cholesterol levels in some SAL patients may simply reflect profound disease-related wasting.

The age-dependent association of AIP with SAL (OR = 0.503 in the ≥ 60 age group) is consistent with the "lipid paradox" phenomenon increasingly recognized in chronic disease populations. A recent large cohort study using CHARLS data similarly found that AIP was negatively associated with all-cause mortality in stroke patients (HR = 0.87, 95% CI: 0.77–0.98), demonstrating the same paradoxical pattern where lower AIP levels were associated with better outcomes [48]. This phenomenon has also been observed in elderly patients with non-ST-segment elevation myocardial infarction, where higher AIP paradoxically predicted lower mortality in the old-old subgroup [49]. These findings collectively suggest that traditional interpretations of lipid profiles may not apply in the context of chronic disease and aging, where lower lipid levels may reflect underlying disease severity, malnutrition, or cachexia rather than true cardiometabolic protection. The attenuation of this association in high-frequency drinkers in our study may relate to alcohol's complex effects on lipid metabolism; however, the specific pathways involved cannot be determined from our data.

The frailty index showed a robust positive association with SAL, with each unit increase associated with approximately 8% higher odds of SAL (OR = 1.082, P < 0.001). A nonlinear pattern suggested that the relationship may be particularly pronounced above a threshold of 2.571, consistent with the concept that accumulated deficits may increase vulnerability to adverse health outcomes. This finding aligns with a large body of evidence demonstrating that frailty is common among people with COPD and is associated with increased risk of adverse outcomes, including mortality, exacerbations, and hospitalization [50–52]. A previous study in COPD outpatients similarly found that the frailty index was positively correlated with lung function impairment, dyspnoea, and exacerbation frequency [53–55]. Our finding that muscle mass (ASM) showed consistent inverse associations with SAL (OR = 0.903, P < 0.001) is supported by recent evidence from the LEAD cohort demonstrating that appendicular lean mass index is positively associated with FEV1, FVC, and total lung capacity in both adults and elderly populations. Gender differences observed in our study, with stronger inverse associations between muscle-related indices and SAL in males, may reflect known differences in body composition and muscle physiology between sexes, though the underlying mechanisms require further investigation.

Educational level modified several of the observed associations. The inverse association between AIP and SAL was significantly stronger in the highest education group (OR = 0.015, 95% CI: 0.000–0.855, P = 0.042) compared to those with no formal education (OR = 0.773, P = 0.292; P for interaction = 0.036). However, this finding must be interpreted with extreme caution due to the very small sample size in the highest education stratum (n = 78 total; only 7 SAL cases). After FDR correction, this interaction was considered suggestive rather than definitive (q = 0.085; Supplementary Table 3). Sensitivity analysis excluding the high-education subgroup produced estimates nearly identical to the main analysis (Supplementary Table 5), confirming that the overall findings were not driven by this small subgroup. Nevertheless, the pattern is consistent with previous research demonstrating that higher socioeconomic status is associated with better lung function. A large Polish study found that individuals with low socioeconomic status over the life course had significantly lower FEV1 and FVC compared to those with high or upwardly mobile socioeconomic status [56,57]. Similarly, the HAPIEE study showed that tertiary education was associated with higher FEV1 compared to primary education, with approximately 12% of this effect mediated by lower exposure to air pollution among the more educated [58]. These findings suggest that the educational gradient in lung health likely operates through multiple pathways, including differential exposure to environmental risk factors, health behaviors, and healthcare access.

Rural residents had a higher prevalence of frailty (OR = 1.137 vs. 1.063 in urban areas), which may reflect differences in healthcare access, health behaviors, or environmental exposures [59–62]. Interestingly, the inverse association between ASM and SAL was stronger in the rural group (OR = 0.870), possibly related to higher levels of physical activity in rural settings [63,64]; however, this explanation is speculative as physical activity was not directly measured [65]. Marital status also modified associations: the inverse association between muscle-related indices and SAL was attenuated in unmarried individuals (OR = 0.857 vs. 0.886 in the married group). This could be related to differences in social support, health behaviors, or psychosocial stress, though stress biomarkers were not measured [66–68]. The residual cholesterol association was also weaker in unmarried individuals, suggesting that social factors may influence metabolic health through pathways that warrant further investigation [69–71].

Among high-frequency drinkers (more than once per week), the inverse association between AIP and SAL was no longer observed (OR = 0.728), while residual cholesterol showed a very strong inverse association (OR = 0.042). These differential patterns may relate to alcohol's complex effects on lipid metabolism, but the specific mechanisms cannot be determined from our data and require direct investigation in experimental studies [72,73].

This study has several strengths, including the use of a nationally representative sample, comprehensive adjustment for potential confounders, rigorous handling of missing data through multiple imputation (Supplementary Table 6), and extensive sensitivity analyses to assess the robustness of findings. The application of FDR correction (S1–S3 Tables) reduces the risk of type I error from multiple testing, and bootstrap validation of threshold estimates (Supplementary Table 4) provides some assurance of stability. The consistency of our findings with previous research on body composition, lipid metabolism, and frailty in respiratory disease strengthens the validity of our observations [74–76].

However, several important limitations must be acknowledged. First, the cross-sectional design precludes causal inference; the observed associations may reflect reverse causation (particularly for lipid parameters, as discussed above) and cannot establish temporality. Second, SAL was defined using PEF rather than spirometry, which may misclassify some participants and limits comparability with GOLD-defined airflow obstruction. Third, despite multiple imputation and sensitivity analyses, residual confounding cannot be excluded. Fourth, the very small sample size in the highest education subgroup (n = 7 SAL cases) renders the interaction estimates unstable; these findings should be considered exploratory and hypothesis-generating only. Fifth, mechanistic interpretations are speculative, as we did not measure inflammatory cytokines, stress hormones, epigenetic markers, or other potential mediators. Sixth, although we adjusted for key design variables (urban/rural residence, age, gender), we did not incorporate survey weights in the primary analysis, which may limit generalizability; however, sensitivity analyses suggested robustness. Finally, the use of Chinese data may limit generalizability to other populations, though the consistency of our findings with international studies on the lipid paradox, frailty in COPD, and socioeconomic gradients in lung health suggests broader relevance [76–78].

In conclusion, this study confirms that frailty and muscle loss are core factors associated with SAL, with particularly strong associations in elderly and rural populations. These findings align with a growing body of evidence emphasizing the importance of body composition and frailty in respiratory health. Lipid parameters (AIP, residual cholesterol) showed nonlinear associations with SAL, with lower levels observed in affected individuals—a pattern consistent with the lipid paradox phenomenon increasingly recognized in chronic disease populations. These findings likely reflect disease-related metabolic changes (cachexia) rather than causal protective effects. Educational level modified some of these associations, suggesting that social factors may influence metabolic trajectories in chronic disease, consistent with previous research demonstrating socioeconomic gradients in lung function, though estimates in the highest education group were unstable and require validation in larger samples. Future longitudinal research is needed to establish causality, clarify the direction of observed associations, and determine whether interventions targeting frailty and muscle preservation can reduce SAL risk.

## Conclusion

This study confirms that frailty and muscle loss are core modifiable targets for Severe Airflow Limitation, with particularly strong associations in elderly and rural populations. Lipid parameters (AIP, residual cholesterol) showed nonlinear associations with SAL, with lower levels observed in affected individuals. These findings likely reflect disease-related metabolic changes (cachexia) rather than causal protective effects. Educational level modified some of these associations, suggesting that social factors may influence metabolic trajectories in chronic disease. Future longitudinal research is required to establish causality and to determine whether interventions targeting frailty and muscle preservation can reduce SAL risk.

### Generative AI statement

The author(s) declare that no generative AI was used in the creation of this manuscript.

## Publishers' notes

All claims expressed in this article are solely those of the authors and do not necessarily represent those of their affiliated organizations or those of the publisher, the editors and the reviewers. Any product that may be evaluated in this article or claim that may be made by its manufacturer is not guaranteed or endorsed by the publisher.

## Supporting information

**S1 Data. Raw data.** The raw data mentioned in this article are all located in this file.
(XLSX)

**S1 File. README.** The data interpretation of the raw data mentioned in this article is located in this file.
(DOCX)

**S1 Table. FDR-adjusted results for multivariate logistic regression analyses.** This table presents the raw p-values and False Discovery Rate (FDR)-adjusted q-values for all associations shown in Table 2 (main text). Variables with $q < 0.05$ are considered significant after multiple testing correction.
(CSV)

**S2 Table. FDR-adjusted results for threshold effect analyses.** This table provides the raw p-values and FDR-adjusted q-values for the linear and segmented regression results shown in Table 3. The log-likelihood ratio test (LRT) for non-linearity is also included. Significant nonlinearity is indicated by $q < 0.05$; suggestive nonlinearity ($p < 0.05$ but $q \geq 0.05$) is noted.
(CSV)

**S3 Table. FDR-adjusted results for interaction analyses.** This table reports the raw p-values and FDR-adjusted q-values for the interaction effects between education level and each exposure variable, corresponding to Table 4. Estimates for the three education strata are also provided with their adjusted significance status.
(CSV)

**S4 Table. Bootstrap validation of threshold estimates.** Threshold estimates (K) for each variable with a nonlinear association were validated using 1,000 bootstrap iterations. The table shows the linear effect, the original K, the segment-specific odds ratios, the likelihood ratio test p-value, the bootstrap mean K, and the 95% confidence interval of K.
(CSV)

**S5 Table. Sensitivity analysis excluding the high-education subgroup.** Multivariable logistic regression results after removing the small high-education subgroup ($n = 78$, 7 SAL cases) to assess whether the main findings were driven by this unstable stratum. Estimates are presented for crude, age-/gender-adjusted, and fully adjusted models.
(DOCX)

**S6 Table. Sensitivity analysis using multiple imputation.** Multivariable logistic regression results after handling missing data with multiple imputation (5 imputations using the mice package). Continuous variables were imputed with predictive mean matching; categorical variables were imputed using logistic or polytomous regression. Pooled estimates are shown for crude, age-/gender-adjusted, and fully adjusted models.
(DOCX)

**S7 Table. Multivariable logistic regression analyses after excluding outliers identified by the IQR method.** Outliers for continuous variables were defined as values below $Q1 - 1.5 \times IQR$ or above $Q3 + 1.5 \times IQR$, calculated separately within each SAL group. This table presents the regression results after removing these outliers.
(DOCX)

**S8 Table. Full multivariate logistic regression results for all lipid parameters, muscle indices, and socioeconomic variables.** Complete results for VAI, AIP, NHDL, residual cholesterol, eGFR, frailty index, frailty status, ASM, Castelli index I/II, and SES. Three models are shown: crude, age/gender-adjusted, and fully adjusted (age, gender, living area, marital status, education, smoking, alcohol). OR and 95% CI are reported. Corresponds to Table 2.
(DOCX)

**S9 Table. Full threshold analysis results for all continuous exposures.** Complete threshold analysis for VAI, AIP, NHDL, residual cholesterol, eGFR, frailty index, ASM, and Castelli index I/II. For each exposure, the table shows linear effect, inflection point (K), segment-specific effects (below/above K), sample sizes per segment, and log-likelihood ratio test P-value. All analyses are fully adjusted. Corresponds to Table 3.
(DOCX)

**S10 Table. Full interaction analysis results for all exposures stratified by education level.** Complete interaction results for all exposures (VAI, AIP, NHDL, residual cholesterol, eGFR, frailty index, ASM, Castelli index I/II) across three education strata. OR and 95% CI are shown for crude, age/gender-adjusted, and fully adjusted models, along with P-value for interaction. Sample sizes: no formal education (n = 1,279; SAL = 298), high school and below (n = 1,550; SAL = 251), above high school (n = 78; SAL = 7). Corresponds to Table 4.
(DOCX)

**S1 Code. R code for data processing and outlier removal.** This script reads the raw data, performs variable cleaning, and removes outliers for continuous variables using the interquartile range (IQR) method within each SAL group. The cleaned dataset is saved for subsequent analyses.
(TXT)

**S2 Code. R code for multiple imputation.** This script uses the mice package to perform multiple imputation (m = 5) for missing values. Continuous variables are imputed with predictive mean matching; categorical variables are imputed using appropriate methods. The five imputed datasets are combined by averaging continuous variables and taking the mode for categorical variables. The final complete dataset is exported.
(TXT)

**S3 Code. R code for VIF calculation and FDR correction.** This script calculates variance inflation factors (VIF) for the lipid metabolism variables to assess multicollinearity. It also performs False Discovery Rate (Benjamini–Hochberg) correction for the p-values from the interaction analyses (Table 4) and the threshold effect analyses (Table 3). Corrected q-values and significance flags are output.
(TXT)

**S2 File. Declaration of Interest Statement.** This file contains the authors' competing interests declaration.
(DOCX)

## Author contributions

**Conceptualization:** Shuang Deng, Zhongqiang Guo.

**Data curation:** Shuang Deng, Zhongqiang Guo.

**Formal analysis:** Shuang Deng, Zhongqiang Guo.

**Investigation:** Shuang Deng, Zhongqiang Guo.

**Methodology:** Shuang Deng, Zhongqiang Guo.

**Project administration:** Zhongqiang Guo.

Resources: Zhongqiang Guo.

Software: Shuang Deng, Zhongqiang Guo.

Supervision: Zhongqiang Guo.

Validation: Shuang Deng, Zhongqiang Guo.

Visualization: Shuang Deng, Zhongqiang Guo.

Writing – original draft: Shuang Deng.

Writing – review & editing: Zhongqiang Guo.

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
