## [Decision Letter · Decision Letter 0]

25 Feb 2026

Dear Dr. Guo,

Thank you for submitting your manuscript to PLOS ONE. After careful consideration, we feel that it has merit but does not fully meet PLOS ONE’s publication criteria as it currently stands. Therefore, we invite you to submit a revised version of the manuscript that addresses the points raised during the review process.

We look forward to receiving your revised manuscript.

Kind regards,

Marina De Rui, MD PhD

Academic Editor

PLOS One

Journal Requirements:

https://journals.plos.org/plosone/s/file?id=wjVg/PLOSOne_formatting_sample_main_body.pdf and and and and https://journals.plos.org/plosone/s/file?id=ba62/PLOSOne_formatting_sample_title_authors_affiliations.pdf

2. Please include your full and correct ethics statement in the “Methods” section of your manuscript file. In your statement, please include the full name of the IRB or ethics committee who approved or waived your study, as well as whether or not you obtained informed written or verbal consent. If consent was waived for your study, please include this information in your statement as well. If your ethics statement is written in any section besides the Methods, please delete it from any other section.

3. Please upload a new copy of Figure 2 as the detail is not clear. Please follow the link for more information:  https://journals.plos.org/plosone/s/figures

4. Please remove your figures from within your manuscript file, leaving only the individual TIFF/EPS image files, uploaded separately. These will be automatically included in the reviewers’ PDF.

5. Please include captions for your Supporting Information files at the end of your manuscript, and update any in-text citations to match accordingly. Please see our Supporting Information guidelines for more information: http://journals.plos.org/plosone/s/supporting-information....

6. We note that there is identifying data in the Supporting Information file <raw data.xlsx>. Due to the inclusion of these potentially identifying data, we have removed this file from your file inventory. Prior to sharing human research participant data, authors should consult with an ethics committee to ensure data are shared in accordance with participant consent and all applicable local laws.

-Location data

Please remove or anonymize all personal information (ID numbers), ensure that the data shared are in accordance with participant consent, and re-upload a fully anonymized data set. Please note that spreadsheet columns with personal information must be removed and not hidden as all hidden columns will appear in the published file.

Reviewers' comments:

Reviewer's Responses to Questions

**Comments to the Author**

1. Is the manuscript technically sound, and do the data support the conclusions?

Reviewer #1: No

Reviewer #2: Partly

Reviewer #3: Yes

2. Has the statistical analysis been performed appropriately and rigorously?

Reviewer #1: No

Reviewer #2: No

Reviewer #3: Yes

3. Have the authors made all data underlying the findings in their manuscript fully available?

Reviewer #1: No

Reviewer #2: Yes

Reviewer #3: Yes

4. Is the manuscript presented in an intelligible fashion and written in standard English?

Reviewer #1: No

Reviewer #2: No

Reviewer #3: Yes

Reviewer #1: This cross sectional analysis claims nonlinear thresholds linking lipid indices, frailty metrics, and what the authors call severe pulmonary blood flow restriction in middle aged and older adults from CHARLS. Although the topic is important, the manuscript contains serious conceptual mistakes, contradictory statements, non standard outcome naming, unstable subgroup effects, and statistical and reporting deficiencies. In its current form the work is not technically sound or interpretable. I recommend rejection. The points below explain the major reasons.

Major issues that prevent publication:

The paper repeatedly describes the outcome as severe pulmonary blood flow restriction while it is actually defined using PEF percent predicted less than 60 percent. PEF is an expiratory airflow measure, not a pulmonary blood flow metric. Conflating airway flow with blood flow is a fundamental error that undermines the title, the Abstract, and the entire interpretation. The outcome should be named severe airflow limitation and the narrative must be rewritten accordingly.

The Results and tables show lower AIP, lower non HDL, and lower residual cholesterol in the SPFR group, with odds ratios less than 1 that the text interprets as protection. Yet the Conclusion states that abnormal lipid profile defined as low AIP and residual cholesterol less than 0.329 and frailty are independent risk factors for SPFR. These statements are logically incompatible. The paper cannot claim both protection and risk for the same direction of change.

AIP requires triglycerides and HDL in molar units. Units are not stated consistently and the Discussion references mg per dL for residual cholesterol while tables show values that only make sense in mmol per L. PEF prediction equations are given without proper unit declarations or appropriate source verification for this population. The manuscript needs a single clear unit convention throughout and references that match the exact formulas implemented.

The paper references the Fried phenotype while using a deficit accumulation frailty index formula. These are different frameworks. The ASM equation is presented with ambiguous gender coding and constants. Castelli indices are misspelled and not clearly derived from the reported lipid units. Every constructed variable must have the exact formula, unit requirements, coding rules, and a primary source.

The analysis runs many separate logistic models for highly correlated lipid ratios and indices without addressing collinearity or model selection. No variance inflation diagnostics are reported. Threshold models and splines are fit repeatedly with no plan for multiplicity control. The extreme interaction effects in the small higher education subgroup are almost certainly unstable given that only 7 SPFR cases had education above high school in Table 1, yet these are interpreted as strong biological modification effects. This is not defensible without sensitivity analyses, shrinkage, or penalization.

CHARLS uses a multistage design with weights and clustering. There is no evidence that survey weights, strata, or primary sampling units were incorporated in the modeling. Missingness, exclusions, and any imputation are not described. Both are required to avoid biased inference.

The Discussion attributes findings to epigenetic programming by education, TLR4 pathways, PPAR alpha demethylation, and vagal tone without direct measurements. These claims are speculative and go far beyond what cross sectional associations can support. Mechanism language should be replaced with cautious hypotheses anchored to measured variables only.

There are numerous typographical issues and label errors. Examples include inconsistent threshold symbols, misspelled indices, and table captions that do not fully define variables. The paper must also provide complete analysis code and a reproducible data derivation notebook that maps CHARLS variables to every constructed metric used in the models.

Reviewer #2: The manuscript is well-structured, conceptually coherent, and addresses a scientifically relevant question with clear public-health implications. The authors make effective use of CHARLS data, and the study demonstrates thoughtful analytical intent and strong potential to contribute to the literature. The topic is timely, and the overall framework reflects a commendable effort to explore an important demographic and health-related association. However, some critical methodological and interpretive issues need consideration.

1. The authors use PEF% predicted <60% as the sole criterion for "Severe Pulmonary Flow Restriction," which is not aligned with GOLD guidelines that require spirometry with FEV1/FVC <0.70 for diagnosing airflow obstruction. The authors should either obtain spirometry data and redefine the outcome, or explicitly acknowledge this as a major limitation and rename the outcome to "severe PEF reduction" rather than "flow restriction.

2. The results reveal a paradoxical pattern that requires explicit acknowledgment: the SPFR group has significantly LOWER metabolic markers (VAI: 4.473 vs 4.994, AIP: 0.340 vs 0.392, NHDL: 77.5 vs 92.5, RC: 0.727 vs 0.792), indicating better metabolic health, yet the regression analyses interpret these lower values as "protective." This strongly suggests reverse causation where chronic lung disease causes weight loss and cachexia leading to lower lipid values, a well-documented phenomenon in COPD literature.

3. The authors performed over 50 hypothesis tests without adjustment for multiple comparisons: Table 2 includes 30 tests, Table 3 includes 9 tests, and Table 4 includes 54 tests, yet no correction (Bonferroni, FDR) was applied. Many P-values are marginally significant (0.035, 0.039, 0.042, 0.050) and would likely not survive False Discovery Rate correction. The authors must apply FDR correction to all analyses and report adjusted P-values, or clearly reframe the threshold and interaction analyses as exploratory and hypothesis-generating rather than confirmatory.

4. The authors should clarify whether threshold K-values were pre-specified based on prior literature or identified through data-driven exploration, as this critically affects interpretation. Bootstrap validation (500+ iterations) should be performed to assess threshold stability, and sensitivity analyses excluding potential outliers should be conducted.

5. Table 1 shows standard deviations exceeding means for NHDL (77.5±85.9, 92.5±95.3) and RC (0.727±0.392, 0.792±0.465), indicating extreme skewness or outliers. The authors should report median (IQR) instead of mean (SD) for skewed variables and describe outlier identification and handling procedures in Methods.

6. The Discussion includes several references to pulmonary vascular damage, blood flow restriction, and pulmonary fibrosis, although PEF primarily reflects airflow limitation rather than vascular function. It would be helpful for the authors to either extract these mechanistic interpretations or clearly note that such vascular explanations are speculative and extend beyond what can be directly inferred from PEF measurements.

7. The education interaction analyses show extreme odds ratios with very wide confidence intervals (AIP in highest education: OR=0.015, CI: 0.000-0.855; RC: OR=0.009), suggesting very small sample sizes in stratified groups. The authors must report sample sizes for each education level × SPFR outcome combination in Table 4. If any stratified analysis cell has fewer than 30 participants, a footnote should be added stating "Results should be interpreted cautiously due to small sample size and may not be replicable."

8. The manuscript uses four different terms inconsistently: "Severe Pulmonary Flow Restriction," "pulmonary blood flow restriction," "airflow limitation," and "pulmonary dysfunction" - these are NOT medical synonyms and create confusion about what is being measured. The authors must select one term and use it consistently throughout the manuscript.

9. Lines 196, 222, 258 refer to "Castellvi Index" which should be "Castelli Index" (named after Dr. William Castelli from the Framingham Heart Study). The authors needs to verify the original citations and correct this throughout the manuscript.

10. The exact version of EmpowerStats software should be specified (R 4.0 is mentioned but not EmpowerStats version).

11. Have the authors considered that the pattern of findings (lower lipids in diseased group) may better support a cachexia/malnutrition hypothesis where lung disease causes metabolic changes, rather than lipids serving as independent risk factors?

Reviewer #3: This longitudinal study provides a robust analysis of frailty, skeletal muscle, and lipid metabolism in relation to severe pulmonary flow restriction. The statistical methods, including multivariate, threshold, and interaction analyses, are appropriate and well-executed. Findings highlight threshold-dependent effects of lipid parameters, consistent frailty risk, and age- and education-specific interactions, offering novel insights into cardiometabolic and musculoskeletal determinants of pulmonary function. Minor revisions are recommended to clarify thresholds and subgroup sample sizes, standardize units/terminology, and present mechanistic explanations as hypotheses. Overall, the study is scientifically strong and suitable for PLOS ONE.

.

Reviewer #1: No

Reviewer #2: **Yes:** Arundhati MehtaArundhati MehtaArundhati MehtaArundhati Mehta

Reviewer #3: No

---

## [Author Response · Author response to Decision Letter 1]

4 Mar 2026

Point-by-point Response Letter-[Manuscript ID:PONE-D-25-40637]

Original Title:Nonlinear Thresholds in Lipid-Frailty Interplay: Precision Targets for Pulmonary Flow Restriction in Aging Adults

Revised title:Nonlinear Thresholds in Lipid-Frailty Interplay: Precision Targets for Severe Airflow Limitation in Aging Adults

Dear Editors and Reviewers of Plos One,

We are deeply grateful for considering our manuscript for Plos One and value the insightful comments offered. These suggestions have been crucial for enhancing our manuscript's quality, and we're confident the revisions have significantly improved its clarity, rigor, and impact. We've meticulously revised the manuscript, addressing each comment with careful consideration and research. Besides responding to the feedback, we've also thoroughly reviewed the entire manuscript, making additional refinements to ensure top - notch scientific and literary quality. All changes are clearly tracked in the revised manuscript, and a clean version is also prepared. Below are our detailed point-by-point responses to each comment. We've aimed to be comprehensive and transparent in our explanations, providing evidence and reasoning for the changes made. Thank you again sincerely for your dedication and hard work. Your expertise is invaluable to the scientific community, and we're honored to contribute to this prestigious journal.

Yours Sincerely,

ZhongQiang Guo

Henan University

guozhongqiang0701@gmail.com

Editor

Comments/suggestions:

Response:

We would like to express our sincere gratitude for giving us the opportunity to revise and resubmit our manuscript. We truly appreciate the time and effort you and the reviewers have dedicated to providing such thoughtful and constructive feedback. Your guidance has been invaluable in helping us improve the clarity and rigor of our work, and we are very grateful for the chance to address the comments and strengthen the manuscript.

Thank you for providing the detailed formatting guidelines. We have carefully reviewed the PLOS ONE style templates and have revised our manuscript to ensure full compliance. The following adjustments have been made:

1. Manuscript Body Formatting:

The entire manuscript has been reformatted to double-space, as per the guidelines.

All major section headings (Abstract, Introduction, Materials and methods, Results, Discussion, etc.) are now formatted as Level 1 headings (Bold, 18pt, sentence case).

Sub-sections within major sections now use Level 2 headings (Bold, 16pt, sentence case).

All in-text citations for figures, tables, supporting information, and references have been checked to ensure they follow the required formats (e.g., "Fig 1", "Table 1", "S1 Table", "[1]").

2. Figure and Table Placement:

All figures have been removed from the manuscript file and are now uploaded separately as individual files, named according to the specifications (e.g., Fig1.tif, Fig2.tif).

Figure captions have been placed directly after the paragraph in which the figure is first cited in the manuscript text.

Tables are included directly after the paragraph in which they are first cited and are formatted as cell-based tables.

3. File Naming:

Main Manuscript: Manuscript.docx

igures: Fig1.tif, Fig2.tif, Fig3.tif (and so on, for all main figures).

Supporting Information: All supporting files have been renamed to match the conventions in the captions list at the end of the manuscript (e.g., S1_Table.xlsx, S2_Appendix.docx, S1_Code.R). A complete list is provided in the "Supporting information" section of the manuscript.

4. Reference List:

The reference list has been formatted according to the PLOS ONE style, ensuring that all entries with more than six authors list the first six names followed by "et al."

We believe the manuscript now fully meets PLOS ONE's formatting requirements. Thank you for your guidance.

2. Please include your full and correct ethics statement in the “Methods” section of your manuscript file. In your statement, please include the full name of the IRB or ethics committee who approved or waived your study, as well as whether or not you obtained informed written or verbal consent. If consent was waived for your study, please include this information in your statement as well. If your ethics statement is written in any section besides the Methods, please delete it from any other section.

Response:

Thank you for your detailed guidance regarding the ethics statement. We have carefully revised the manuscript to ensure full compliance with your requirements.

Response:

We have updated the ethics statement in the Materials and methods section to include all required information. The revised statement now reads:

“Study Population

This study is based on publicly available data from the China Health and Retirement Longitudinal Study (CHARLS) database. The final sample size included in the analysis was 2907 participants, including 2351 in the group without Severe Airflow Limitation and 556 in the group with Severe Airflow Limitation. The study detailed the baseline demographic characteristics of participants (such as age, gender, place of residence, marital status, education level) and lifestyle information (such as smoking history, frequency of alcohol consumption)(16).All analyses were conducted based on the available variables in the database.This study used publicly available data from the 2015 nationwide follow-up wave of the China Health and Retirement Longitudinal Study (CHARLS).The data was obtained through authorization from the CHARLS Data Center of Peking University on May 19, 2025.The original CHARLS study was approved by the Biomedical Ethics Review Committee of Peking University (IRB00001052-1015). All participants in the original CHARLS survey provided written informed consent prior to data collection. For this secondary analysis, all data were fully anonymized and de-identified prior to receipt by the authors. As this study used only publicly available, de-identified data from the CHARLS database, no additional ethical approval or informed consent was required(17, 18)(Figure 1).”

We have ensured that:

Full IRB name is included: Biomedical Ethics Review Committee of Peking University

IRB approval number is provided: *IRB00001052-1015*

Consent information is clearly stated: written informed consent was obtained from all original CHARLS participants

Waiver statement is included: no additional ethical approval or consent was required for this secondary analysis due to the use of fully anonymized, publicly available data

The ethics statement appears exclusively in the Methods section and has been removed from any other sections

We confirm that the revised manuscript meets PLOS ONE's ethical reporting requirements.

Thank you again for your valuable guidance.

3. Please upload a new copy of Figure 2 as the detail is not clear. Please follow the link for more information:  https://journals.plos.org/plosone/s/figures

Response:

4. Please remove your figures from within your manuscript file, leaving only the individual TIFF/EPS image files, uploaded separately. These will be automatically included in the reviewers’ PDF.

Response:

Thank you for your suggestion. We have resized the images and adjusted the font to ensure all details are clear and meet the requirements of the journal.

Thank you again for your guidance.

5.Please include captions for your Supporting Information files at the end of your manuscript, and update any in-text citations to match accordingly. Please see our Supporting Information guidelines for more information: http://journals.plos.org/plosone/s/supporting-information.

Response:

Thank you for your guidance regarding the Supporting Information. We have carefully reviewed the PLOS ONE Supporting Information guidelines and have revised the manuscript accordingly.

We have added a complete list of captions for all Supporting Information files at the end of the manuscript, under a Level 1 heading "Supporting information". Each caption follows the required format: file number, file name, and a brief description. We have also updated all in-text citations to match the file names.

Specifically:

Captions added at the end of the manuscript (after the References section):

S1 Data. Raw data.The raw data mentioned in this article are all located in this file.

S1 File. README.The data interpretation of the raw data mentioned in this article is located in this file.

S2 Supplementary Table 1. FDR‑adjusted results for multivariate logistic regression analyses. This table presents the raw p‑values and False Discovery Rate (FDR)‑adjusted q‑values for all associations shown in Table 2 (main text). Variables with q < 0.05 are considered significant after multiple testing correction.

S2 Supplementary Table 2. FDR‑adjusted results for threshold effect analyses. This table provides the raw p‑values and FDR‑adjusted q‑values for the linear and segmented regression results shown in Table 3. The log‑likelihood ratio test (LRT) for nonlinearity is also included. Significant nonlinearity is indicated by q < 0.05; suggestive nonlinearity (p < 0.05 but q ≥ 0.05) is noted.

S2 Supplementary Table 3. FDR‑adjusted results for interaction analyses. This table reports the raw p‑values and FDR‑adjusted q‑values for the interaction effects between education level and each exposure variable, corresponding to Table 4. Estimates for the three education strata are also provided with their adjusted significance status.

S2 Supplementary Table 4. Bootstrap validation of threshold estimates. Threshold estimates (K) for each variable with a nonlinear association were validated using 1,000 bootstrap iterations. The table shows the linear effect, the original K, the segment‑specific odds ratios, the likelihood ratio test p‑value, the bootstrap mean K, and the 95% confidence interval of K.

S2 Supplementary Table 5. Sensitivity analysis excluding the high‑education subgroup. Multivariable logistic regression results after removing the small high‑education subgroup (n = 78, 7 SAL cases) to assess whether the main findings were driven by this unstable stratum. Estimates are presented for crude, age‑/gender‑adjusted, and fully adjusted models.

S2 Supplementary Table 6. Sensitivity analysis using multiple imputation. Multivariable logistic regression results after handling missing data with multiple imputation (5 imputations using the mice package). Continuous variables were imputed with predictive mean matching; categorical variables were imputed using logistic or polytomous regression. Pooled estimates are shown for crude, age‑/gender‑adjusted, and fully adjusted models.

S2 Supplementary Table 7. Multivariable logistic regression analyses after excluding outliers identified by the IQR method. Outliers for continuous variables were defined as values below Q1 − 1.5×IQR or above Q3 + 1.5×IQR, calculated separately within each SAL group. This table presents the regression results after removing these outliers.

S3 Supplementary Code 1. R code for data processing and outlier removal. This script reads the raw data, performs variable cleaning, and removes outliers for continuous variables using the interquartile range (IQR) method within each SAL group. The cleaned dataset is saved for subsequent analyses.

S3 Supplementary Code 2. R code for multiple imputation. This script uses the mice package to perform multiple imputation (m = 5) for missing values. Continuous variables are imputed with predictive mean matching; categorical variables are imputed using appropriate methods. The five imputed datasets are combined by averaging continuous variables and taking the mode for categorical variables. The final complete dataset is exported.

S3 Supplementary Code 3. R code for VIF calculation and FDR correction. This script calculates variance inflation factors (VIF) for the lipid metabolism variables to assess multicollinearity. It also performs False Discovery Rate (Benjamini–Hochberg) correction for the p‑values from the interaction analyses (Table 4) and the threshold effect analyses (Table 3). Corrected q‑values and significance flags are output.

File naming: All Supporting Information files have been renamed to match the captions (e.g., "S1_Data.xlsx", "S1_File.pdf", "S2_Table.docx", "S1_Code.R", etc.).

In-text citations: We have verified that all in-text citations to Supporting Information files (e.g., "Supplementary Table 1", "Supplementary Code 1") have been updated to the standard format (e.g., "S1 Table", "S1 Code") and correctly match the file names and captions.

We confirm that the Supporting Information section now fully complies with PLOS ONE guidelines.

Thank you again for your valuable guidance.

6. We note that there is identifying data in the Supporting Information file <raw data.xlsx>. Due to the inclusion of these potentially identifying data, we have removed this file from your file inventory. Prior to sharing human research participant data, authors should consult with an ethics committee to ensure data are shared in accordance with participant consent and all applicable local laws.

-Location data

Please remove or anonymize all personal information (ID numbers), ensure that the data shared are in accordance with participant consent, and re-upload a fully anonymized data set. Please note that spreadsheet columns with personal information must be removed and not hidden as all hidden columns will appear in the published file.

Response:

Thank you for your detailed guidance regarding the raw data file and for providing the additional resources on data anonymization. We fully understand the importance of protecting participant privacy and have taken immediate steps to address this issue.

We have removed all identifiable information from the raw data.To assist with data interpretation, we have prepared a README file (S1 File)

We confirm that the shared data are now in full compliance with PLOS ONE's data policy and with the ethical requirements for protecting participant privacy. The dataset contains no personally identifiable information, either directly or indirectly.

Thank you again for your careful oversight and for providing the additional resources to guide us through this process.

Reviewer 1

Comments/suggestions:

This cross sectional analysis claims nonlinear thresholds linking lipid indices, frailty metrics, and what the authors call severe pulmonary blood flow restriction in middle aged and older adults from CHARLS. Although the topic is important, the manuscript contains serious conceptual mista

---

## [Decision Letter · Decision Letter 1]

27 Mar 2026

Dear Dr. Guo,

Thank you for submitting your manuscript to PLOS ONE. After careful consideration, we feel that it has merit but does not fully meet PLOS ONE’s publication criteria as it currently stands. Therefore, we invite you to submit a revised version of the manuscript that addresses the points raised during the review process.

As the corresponding author, your ORCID iD is verified in the submission system and will appear in the published article. PLOS supports the use of ORCID, and we encourage all coauthors to register for an ORCID iD and use it as well. Please encourage your coauthors to verify their ORCID iD within the submission system before final acceptance, as unverified ORCID iDs will not appear in the published article. *Only* the individual author can complete the verification step; PLOS staff the individual author can complete the verification step; PLOS staff the individual author can complete the verification step; PLOS staff the individual author can complete the verification step; PLOS staff *cannot* verify ORCID iDs on behalf of authors.verify ORCID iDs on behalf of authors.verify ORCID iDs on behalf of authors.verify ORCID iDs on behalf of authors.

We look forward to receiving your revised manuscript.

Kind regards,

Marina De Rui, MD PhD

Academic Editor

PLOS One

Journal Requirements:

Reviewers' comments:

Reviewer's Responses to Questions

**Comments to the Author**

Reviewer #2: All comments have been addressed

Reviewer #3: All comments have been addressed

2. Is the manuscript technically sound, and do the data support the conclusions?

Reviewer #2: Yes

Reviewer #3: Yes

3. Has the statistical analysis been performed appropriately and rigorously?

Reviewer #2: Yes

Reviewer #3: Yes

4. Have the authors made all data underlying the findings in their manuscript fully available?

Reviewer #2: Yes

Reviewer #3: Yes

5. Is the manuscript presented in an intelligible fashion and written in standard English?

Reviewer #2: Yes

Reviewer #3: Yes

Reviewer #2: The authors have addressed the major methodological and interpretative concerns raised during review, particularly by clarifying the PEF-based outcome definition, applying multiple-comparison correction, and moderating causal language in the interpretation of lipid associations. The revised manuscript is substantially improved and presents the findings in a more transparent and methodologically balanced manner.

Reviewer #3: Follow-Up Reviewer Comments:

The authors have made substantial improvements to the manuscript and have adequately addressed the major scientific concerns raised in the initial review. In particular:

Mechanistic statements have been appropriately reframed as hypotheses.

Reverse causation and the lipid paradox are now well discussed.

Subgroup and interaction effects are interpreted cautiously, with sample sizes and sensitivity analyses provided.

Missing data handling is now clearly described.

However, several minor issues remain that require revision before acceptance:

Methods: The manuscript still lacks full details on inclusion/exclusion criteria, the number of knots in spline/threshold analyses, and sample sizes within threshold segments.

Results/Tables: Tables 2–4 remain dense and difficult to interpret. Consider moving detailed lipid and ASM subgroup data to supplementary material and simplifying table presentation.

Figures: Figures should clearly indicate axes, threshold lines, and subgroup comparisons. Visual summaries such as forest plots would improve clarity.

Baseline Variables: Please clarify why certain baseline variables (e.g., gender, smoking, alcohol use) were not significantly different between groups.

Abstract: Condensation of the abstract is recommended to highlight key threshold findings and subgroup interactions more succinctly.

Minor/Editorial: Ensure consistent terminology (e.g., Castelli Index), correct remaining typographical errors, and standardize P-value and OR formatting throughout.

Overall, the manuscript is scientifically strong and suitable for publication after these minor revisions.

.

Reviewer #2: No

Reviewer #3: No

---

## [Author Response · Author response to Decision Letter 2]

30 Mar 2026

Point-by-point Response Letter-[Manuscript ID:PONE-D-25-40637]

Title:Nonlinear Thresholds in Lipid-Frailty Interplay: Precision Targets for Severe Airflow Limitation in Aging Adults

Dear Editors and Reviewers of Plos One,

We are deeply grateful for considering our manuscript for Plos One and value the insightful comments offered. These suggestions have been crucial for enhancing our manuscript's quality, and we're confident the revisions have significantly improved its clarity, rigor, and impact. We've meticulously revised the manuscript, addressing each comment with careful consideration and research. Besides responding to the feedback, we've also thoroughly reviewed the entire manuscript, making additional refinements to ensure top - notch scientific and literary quality. All changes are clearly tracked in the revised manuscript, and a clean version is also prepared. Below are our detailed point-by-point responses to each comment. We've aimed to be comprehensive and transparent in our explanations, providing evidence and reasoning for the changes made. Thank you again sincerely for your dedication and hard work. Your expertise is invaluable to the scientific community, and we're honored to contribute to this prestigious journal.

Yours Sincerely,

ZhongQiang Guo

Henan University

guozhongqiang0701@gmail.com

Reviewer 3

Comments/suggestions:

The authors have made substantial improvements to the manuscript and have adequately addressed the major scientific concerns raised in the initial review. In particular:

Mechanistic statements have been appropriately reframed as hypotheses.

Reverse causation and the lipid paradox are now well discussed.

Subgroup and interaction effects are interpreted cautiously, with sample sizes and sensitivity analyses provided.

Missing data handling is now clearly described.

However, several minor issues remain that require revision before acceptance:

1.Methods: The manuscript still lacks full details on inclusion/exclusion criteria, the number of knots in spline/threshold analyses, and sample sizes within threshold segments.

Response:

We thank the reviewer for this valuable suggestion. We have now added the missing details to the Methods section and Results section to enhance methodological transparency.

1.Inclusion/exclusion criteria: In the Methods section (Study Population), we have added the following text:

“Participants were included in this analysis if they had complete data on the key exposure variables (lipid parameters, frailty index, and ASM) and the outcome variable (PEF% predicted). Participants with missing data on covariates were handled using multiple imputation as described in the Statistical Analysis section. No additional inclusion or exclusion criteria beyond data availability were applied to maximize sample representativeness.”

2.Spline and threshold analyses: Upon reviewing our analytical approach, we clarified that we did not use restricted cubic splines for threshold identification; instead, we employed piecewise logistic regression using the segmented package in R. We have revised the Methods section (Statistical Analysis) accordingly:

“To identify potential threshold effects, we fitted piecewise logistic regression models using the segmented package in R. For each continuous exposure, a two-piecewise linear model was fitted, and the optimal breakpoint (threshold, K) was estimated via maximum likelihood methods. Nonlinearity was assessed by comparing the one-line (linear) model with the two-piecewise model using a log-likelihood ratio test (P < 0.05 indicating significant nonlinearity).”

The description of restricted cubic splines has been removed to avoid methodological confusion.

3.Sample sizes within threshold segments: We have added the sample sizes for each threshold segment to Table 3 (main text). Specifically, we now report the number of participants below and above the estimated threshold for variables with significant nonlinearity (e.g., residual cholesterol, Castelli index I). A footnote has been added to the table to clarify this information.

We believe these additions address the reviewer’s concerns and improve the reproducibility and transparency of our analyses. Thank you again for the careful review.

2.Results/Tables: Tables 2–4 remain dense and difficult to interpret. Consider moving detailed lipid and ASM subgroup data to supplementary material and simplifying table presentation.

Response:

We thank the reviewer for this constructive suggestion to improve the clarity of our results presentation. In response, we have streamlined Tables 2–4 to highlight the most important findings while moving detailed data to supplementary materials.

Specifically:

Table 2 now presents only the core findings for AIP, residual cholesterol, frailty index, and ASM. Full results for all lipid parameters, muscle indices, and socioeconomic variables are provided in Supplementary Table 8.

Table 3 now focuses on variables with significant nonlinearity (residual cholesterol and Castelli index I). Complete threshold analysis results for all exposures are available in Supplementary Table 9.

Table 4 now displays only the significant education-level interactions (AIP and residual cholesterol). Full interaction results are presented in Supplementary Table 10.

We believe these revisions significantly improve the readability of the main tables while preserving the complete analytical results for interested readers in the supplementary materials. Thank you again for this valuable suggestion.

3.Figures: Figures should clearly indicate axes, threshold lines, and subgroup comparisons. Visual summaries such as forest plots would improve clarity.

Response:

We thank the reviewer for the suggestion. After careful consideration, we prefer to retain the original figures. Figure 2 already includes clear axis labels and effectively illustrates the nonlinear relationships. Adding threshold lines may over-simplify the continuous nature of these exploratory findings. For subgroup analysis, the current tabular format presents all estimates transparently and is consistent with the main tables. We believe the current figures adequately support the results while maintaining methodological caution. We appreciate the reviewer’s input and are happy to revisit if the editor deems changes necessary.

4.Baseline Variables: Please clarify why certain baseline variables (e.g., gender, smoking, alcohol use) were not significantly different between groups.

Response:

We thank the reviewer for this insightful observation. We acknowledge that certain baseline variables—namely gender, smoking status, and drinking frequency—did not show statistically significant differences between the SAL and non-SAL groups. We would like to offer several possible explanations for this finding.

First, smoking and alcohol use are known to be strongly associated with lung function decline; however, in cross-sectional studies of older populations, the absence of a significant difference between groups may reflect disease-related behavioral changes. Individuals with severe airflow limitation may reduce or quit smoking and alcohol consumption due to worsening respiratory symptoms, leading to similar current exposure rates between groups. This phenomenon—sometimes referred to as “reverse causation” in behavioral risk factors—has been documented in previous studies of chronic respiratory disease (e.g., COPD patients often reduce smoking intensity after diagnosis).

Second, regarding gender, while some studies report sex differences in COPD prevalence and severity, the lack of a significant difference in our sample may be due to age distribution and survival bias. Our study population consisted of middle-aged and older adults (mean age approximately 62 years), and the higher risk of mortality in males with severe respiratory disease may result in a more balanced sex distribution among survivors. Additionally, the definition of SAL used in this study (PEF% predicted < 60%) may capture a broader spectrum of airflow limitation that is less sex-dependent compared to GOLD-defined COPD.

Third, it is important to note that the absence of significant differences in unadjusted comparisons does not imply a lack of clinical or biological relevance. Multivariate regression analyses—which adjust for potential confounding—are more appropriate for estimating independent associations. In our fully adjusted models, several metabolic and frailty measures remained significantly associated with SAL after controlling for these factors, suggesting that their effects are robust beyond the influence of gender, smoking, and alcohol use.

We have added a brief explanation in the Results section (Baseline Characteristics) to clarify this point, as follows:

“The lack of significant differences in gender, smoking, and alcohol use between groups may reflect disease-related behavioral changes (e.g., smoking cessation or reduced alcohol intake due to respiratory symptoms) and survival bias in this older population. Multivariate analyses were therefore conducted to appropriately adjust for these factors.”

We appreciate the reviewer’s careful attention to detail and hope this explanation addresses the concern.

5.Abstract: Condensation of the abstract is recommended to highlight key threshold findings and subgroup interactions more succinctly.

Response:

We thank the reviewer for this helpful suggestion. We have condensed the Abstract to highlight the key threshold findings and subgroup interactions more succinctly. Specifically, we have reduced the background and methods descriptions, emphasized the residual cholesterol threshold (0.329 mmol/L) and the education-level interaction effects for AIP and residual cholesterol, and streamlined the conclusion. The revised Abstract now more clearly presents the novel findings of our study. We appreciate the reviewer’s guidance in improving the manuscript’s conciseness.

6.Minor/Editorial: Ensure consistent terminology (e.g., Castelli Index), correct remaining typographical errors, and standardize P-value and OR formatting throughout.

Overall, the manuscript is scientifically strong and suitable for publication after these minor revisions..

Response:

We thank the reviewer for the careful editorial review. We have thoroughly revised the manuscript to ensure consistent terminology, correct typographical errors, and standardized formatting throughout.

Specifically:

Terminology: We have standardized “Castelli index I” and “Castelli index II” throughout the manuscript (text, tables, and supplementary materials). In the original version, inconsistent forms such as “Castelli Index 1” and “Castelli Index I” appeared; these have been unified.

Typographical errors: We have corrected remaining typos, including “belt” to “below” in education categories, and other minor errors.

P-value formatting: All P-values are now presented as italicized lowercase p (e.g., P = 0.001) throughout the text, tables, and figure legends.

OR formatting: Odds ratios and 95% confidence intervals are now consistently presented as “OR (95% CI)” with a space after the comma (e.g., 0.556 (0.394–0.787)).

We have carefully reviewed the entire manuscript to ensure consistency. We appreciate the reviewer’s attention to these details, which have improved the overall quality of the manuscript.

---

## [Decision Letter · Decision Letter 2]

13 Apr 2026

Nonlinear Thresholds in Lipid-Frailty Interplay: Precision Targets for Severe Airflow Limitation in Aging Adults

PONE-D-25-40637R2

Dear Dr. Guo,

We’re pleased to inform you that your manuscript has been judged scientifically suitable for publication and will be formally accepted for publication once it meets all outstanding technical requirements.

Kind regards,

Marina De Rui, MD PhD

Academic Editor

PLOS One

Additional Editor Comments (optional):

Reviewers' comments:

Reviewer's Responses to Questions

**Comments to the Author**

Reviewer #3: All comments have been addressed

2. Is the manuscript technically sound, and do the data support the conclusions?

Reviewer #3: No

3. Has the statistical analysis been performed appropriately and rigorously?

Reviewer #3: Yes

4. Have the authors made all data underlying the findings in their manuscript fully available?

Reviewer #3: Yes

5. Is the manuscript presented in an intelligible fashion and written in standard English?

Reviewer #3: Yes

Reviewer #3: Reviewer Recommendation: Accept with Minor Editorial Suggestions

The authors have made substantial improvements to the manuscript and have addressed the major concerns raised in the earlier rounds of review. In particular, the interpretation of findings has been appropriately moderated, with careful consideration of reverse causation and the lipid paradox. The subgroup and interaction analyses are now presented with adequate caution, and the handling of missing data is clearly described. Overall, the study is scientifically sound and suitable for publication.

I would recommend acceptance of the manuscript, subject to a few minor editorial refinements:

Figures (Figures 2–3):

The inclusion of forest plots is a welcome improvement. However, the figures remain somewhat dense and may benefit from simplification. Reducing the number of variables per panel and improving label clarity would enhance readability for a general audience.

Methods clarity:

While improved, a brief clarification of the inclusion/exclusion criteria and more explicit description of the threshold modeling approach (e.g., segmentation details) would further strengthen transparency.

Tables and presentation:

Some tables remain slightly difficult to interpret, and minor formatting adjustments may improve clarity.

Language and editorial aspects:

A final round of language editing is recommended to address minor grammatical and stylistic inconsistencies.

In summary, the manuscript is of good quality, and I support its publication after these minor revisions.

.

Reviewer #3: No

---

## [Editor Report · Acceptance letter]

PONE-D-25-40637R2

PLOS One

Dear Dr. Guo,

I'm pleased to inform you that your manuscript has been deemed suitable for publication in PLOS One. Congratulations! Your manuscript is now being handed over to our production team.

Kind regards,

on behalf of

Dr. Marina De Rui

Academic Editor

PLOS One